# Nst1, Densely Associated to P-Body in the Post-Exponential Phases of *Saccharomyces cerevisiae*, Shows an Intrinsic Potential of Producing Liquid-Like Condensates of P-Body Components in Cells

**DOI:** 10.3390/ijms23052501

**Published:** 2022-02-24

**Authors:** Yoon-Jeong Choi, Kiwon Song

**Affiliations:** Department of Biochemistry, College of Life Science and Biotechnology, Yonsei University, Seoul 03722, Korea; yoon3004@yonsei.ac.kr

**Keywords:** Nst1, P-body, Edc3, mRNP granule, condensation, liquid–liquid phase separation, post-exponential growth phases

## Abstract

Membrane-less biomolecular compartmentalization is a core phenomenon involved in many physiological activities that occur ubiquitously in cells. Condensates, such as promyelocytic leukemia (PML) bodies, stress granules, and P-bodies (PBs), have been investigated to understand the process of membrane-less cellular compartmentalization. In budding yeast, PBs dispersed in the cytoplasm of exponentially growing cells rapidly accumulate in response to various stresses such as osmotic stress, glucose deficiency, and heat stress. In addition, cells start to accumulate PBs chronically in post-exponential phases. Specific protein–protein interactions are involved in accelerating PB accumulation in each circumstance, and discovering the regulatory mechanism for each is the key to understanding cellular condensation. Here, we demonstrate that Nst1 of budding yeast *Saccharomyces cerevisiae* is far more densely associated with PBs in post-exponentially growing phases from the diauxic shift to the stationary phase than during glucose deprivation of exponentially growing cells, while the PB marker Dcp2 exhibits a similar degree of condensation under these conditions. Similar to Edc3, ectopic Nst1 overexpression induces self-condensation and the condensation of other PB components, such as Dcp2 and Dhh1, which exhibit liquid-like properties. Altogether, these results suggest that Nst1 has the intrinsic potential for self-condensation and the condensation of other PB components, specifically in post-exponential phases.

## 1. Introduction

Understanding the mechanism of intracellular molecular compartmentalization is essential to understanding the mechanics of life. The membrane-less compartmentalization of biomolecules by liquid–liquid phase separation (LLPS) has been reported ubiquitously in cells [1,2]: Cajal bodies, nuclear speckles, paraspeckles, histone-locus bodies, nuclear gems, promyelocytic leukemia (PML) nuclear bodies (NBs) in the nucleus and processing(P)-bodies (PBs), stress granules (SGs), and germ granules in the cytoplasm.

Each condensate contains a unique group of proteins and RNA/DNA, which differentiates it from the others [3,4,5,6,7]. Not all associated components, but a few molecular factors, are reported to be required to form condensates. Specific proteins such as PML (TRIM19) of PML NB, SFPQ of paraspeckles, COILIN of Cajal bodies, G3BP of SGs, and DDX6/LSM14A of PBs in mammalian cells are known representatives of the essential proteins for condensation to form membrane-less organelles [8]. Banani et al., designated these essential molecular components with the potential to induce condensation as ‘scaffolds’ and the other constituents as ‘clients’ [9]. Scaffold proteins have the intrinsic potential to induce phase separation, and client proteins associated with condensates cannot actively induce condensation [9]. Sanders et al., reported that proteins with high valence nodes in protein–protein interaction networks are crucial in generating condensation [8]. These observations indicate the presence of specific key molecules that induce condensation in each condensate.

A PB is a representative messenger ribonucleoprotein (mRNP) granule that forms a dynamically concentrated and dispersed membrane-less cellular structure conserved in eukaryotic cells. The PBs of the budding yeast *Saccharomyces cerevisiae* contain the de-capping enzymes Dcp1 and Dcp2 and the Lsm1–7 complex, as well as Edc3, Pat1, and Dhh1, all of which are mRNA-binding proteins that stimulate mRNA decay [10,11,12]. Similar to other condensates, there are indispensable factors that act as scaffolds for PB accumulation. The accumulated PBs in vivo are dispersed in cycloheximide (CHX)-treated cells, and the purified PBs in vitro are dispersed when treated with RNase, indicating that polysome-free mRNA is a key molecule to induce the assembly of PBs [13]. Edc3 and Lsm4 are also indispensable factors for PB assembly via multivalent interactions, which are driven by the poly Q prion-like domain under glucose-deficient stress and arsenite exposure [13,14]. The ∆*edc3 lsm4*∆*C* double deletion strain displays a severe defect in PB formation during glucose depletion [14]. However, the ∆*edc3 lsm4*∆*C* strain assembles PBs in the stationary phase, demonstrating the presence of additional scaffolds and interactions that promote PB assembly [15]. These findings strongly suggest that PB formation in post-exponential growth phases can be induced by multiple redundant pathways consisting of molecular interactions different from those in the PB formation of stressed mitotic cells. Considering that a specific subset of interactions can predominate to accumulate PBs under specific conditions, discovering novel components of each specific stress condition could be important for understanding non-membranous molecular compartmentalization under diverse physiological conditions.

Nst1 is named based on the phenotype of its knockout mutant, which negatively affects salt tolerance [16]. Goossens et al., first reported that Nst1 physically interacts with the mRNA splicing factor Msl1 in a yeast two-hybrid system and showed growth defects in response to salt stress when deleted [16]. An early study suggested that the physiological function of Nst1 might be involved in stress-responsive signaling pathways. Indeed, we identified the physiological function of Nst1 as a scaffold-like protein cross-linking the Hog pathway and cell wall integrity pathway in response to heat stress, osmotic stress, and pheromones [17]. When examining changes in endogenous Nst1-EGFP localization depending on various stresses, we unexpectedly discovered that Nst1 is dispersed in the cytoplasm during exponential growth but switches to condensates in stationary phase cells.

A genome-wide in vivo protein–protein interaction (PPI) study detected Nst1 as a component of the conserved Ccr4-Not eukaryotic deadenylase complex due to its functional interaction with Caf40, Not3, and Pop2 [18]. Considering that the Ccr4-Not complex is associated with PBs, the reported functional interactions of Nst1 imply that Nst1 is possibly associated with PBs.

Here, we demonstrate that Nst1 is strongly associated with PBs, specifically in post-exponential growth phases, and has the intrinsic potential of oligomerization to induce self-condensation and PB accumulation.

## 2. Results

### 2.1. Nst1 Is Densely Associated with PBs of S. cerevisiae Cells in Post-Exponential Growth Phases

A genome-wide in vivo PPI study by Tarassov et al., showed that Nst1 interacts functionally with Pop2, Not3, and Caf40, which are the components of the Ccr4-Not complex associated with PBs (Figure 1A) [18]. To confirm the functional association between Nst1 and PBs, we performed deletion mutant suppressor screening for Nst1 overexpression. Since we discovered that the Nst1-overexpressed cells show growth retardation compared to the vector control, we screened the deletion mutants of the genes related to the function of PBs or SGs, which suppressed the growth retardation phenotype of the Nst1-overexpressed cells (Appendix A). The 92 deletion mutants in the yeast knock-out (YKO) library (*Saccharomyces* Genome Deletion Project, http://www-sequence.stanford.edu/group/yeast_deletion_project/, accessed on 7 January 2022), which were categorized as genes related to PBs or SGs in the *S. cerevisiae* genome database (SGD, https://www.yeastgenome.org, accessed on 7 January 2022), were used for the screening (Appendix A). *PSP2*, *LSM12*, *DCS1*, *TIF4631*, and *POP2* were identified as suppressor genes whose deletion suppressed the growth retardation phenotype induced by Nst1 overexpression. Considering that ectopic expression of Psp2 recovers PB assembly in glucose-deprived ∆*edc3 lsm4*∆*C* cells [15], and that Pop2 is a component of the Ccr4-Not deadenylase complex, the functional relevance of Nst1 to PB components was suggested. Based on these functional interactions with the previously-known PB components, we started to investigate the possibility of Nst1 as a PB-associated protein. Because PBs are reported to be highly accumulated in stationary-phase cells [19,20], we examined Nst1 localization in the PBs of the stationary phase. With Nst1-EGFP and Edc3-Redstar2 double-tagged wild-type w303a (YSK3519), we observed that Nst1 was condensed and co-localized with the PB marker protein Edc3 in the stationary phase cells (OD_600_ > 5), as shown in Figure 1A.

Three distinctive growth phases have been reported in the chronological growth of *S. cerevisiae*: the exponentially growing log phase (OD_600_ < 1), growth-retarded diauxic shift (1 < OD_600_ < 5), and the stationary phase (OD_600_ > 5), in which a subpopulation of cells arrests at G0 [21,22,23]. Cells in the diauxic shift proliferate more slowly than exponentially growing cells, but the subpopulation of cells does not enter the G0 phase [21,22]. We monitored PB accumulation through the chronic growing phases of budding yeast using w303a wild-type strains whose Dcp2 (a PB marker) was chromosomally tagged with enhanced green fluorescent protein (EGFP) (YSK3527) and observed that cells in post-exponential growing phases (from the diauxic shift to the stationary phase) accumulate Dcp2-EGFP condensates significantly (data not shown). To investigate the association of Nst1 with PBs during chronic growth and in the stress condition of budding yeast, we tried to verify the localization of Nst1 in actively growing log phase cells (OD_600_ < 0.5), diauxic shift cells (OD_600_ 2–3), and glucose-deprived log phase cells. We also compared its localization with that of the PB marker proteins Dcp2 and Edc3 and the SG marker protein Pab1 (Figure 1B). We constructed w303a wild-type strains in which endogenous Nst1 (YSK3524), Dcp2 (YSK3527), Edc3 (YSK3519), and Pab1 (YSK3525) were tagged with EGFP, and the endogenous expression of EGFP-tagged Nst1, Edc3, Dcp2, and Pab1 was confirmed using Western blot analysis (Appendix A). To observe cells during chronic growth and under stress conditions, these strains were primarily cultured to diauxic shift (OD_600_ 2–3) and sub-cultured in fresh Synthetic-defined (SD) medium (OD_600_ < 0.1) to reach the log phase (OD_600_ < 0.5) and further cultured to re-enter diauxic shift (OD_600_ 2–3). The log phase cells were washed and further incubated with glucose-depleted synthetic medium for 10 min for glucose deprivation.

Nst1-EGFP was dispersed in the cytoplasm in log phase cells and then condensed in the cells on a diauxic shift similar to Dcp2-EGFP and Edc3-EGFP (Figure 1B). In contrast, EGFP-Pab1, a marker of another mRNP granule, SG, did not accumulate in the same phase (Figure 1B). The differential localization and condensation of Nst1 as of Dcp2 and Edc3 suggests the probability of its association with PBs. Notably, Nst1-EGFP was also shown as foci in glucose-deprived log phase cells, but the intensities of the foci were not as strong as those of Nst1-EGFP condensates shown in cells on diauxic shift. On the other hand, the intensities of Dcp2-EGFP and Edc3-EGFP condensates induced by glucose deprivation were similar to those induced in cells on diauxic shift (Figure 1B).

Considering that incubation for 10 min in a glucose-depleted medium might not be sufficient to induce condensation of Nst1-EGFP, we checked Nst1 localization in w303a that endogenously expressed EGFP-tagged Nst1 and mKate2-fused to Dcp2 (YSK3642) with 30 min incubation in a glucose-deficient medium. Exponentially growing YSK3642 cells were transferred to a glucose-depleted or glucose-sufficient medium for 30 min. Nst1-EGFP was condensed and co-localized with Dcp2-mKate2 condensates incubated in a glucose-depleted medium (Appendix A). However, Nst1 puncta accumulation was not as obvious as the Nst1 puncta accumulation observed in the diauxic shift. The extremely low signal intensity of Nst1 condensates in the glucose-depleted stress of exponentially growing cells implies that Nst1 plays a unique role in the accumulation of larger PBs in diauxic shift, which differs from those of Dcp2 and Edc3 (Appendix A).

To confirm that the Nst1 condensates in cells of the diauxic shift are co-localized with PBs, we used a w303a wild-type strain in which Nst1 was tagged with EGFP and Dcp2 was tagged with Redstar2 (YSK3509). In these cells, grown to diauxic shift, as shown in Figure 1B, red Dcp2 puncta were clearly detected in the diauxic shift (Figure 1C). The EGFP-tagged Nst1 in the diauxic shift was observed as clear puncta co-localized with red puncta of Redstar2-tagged Dcp2 (Figure 1C), indicating that the condensates of Nst1 in cells on the diauxic shift were localized in PBs.

### 2.2. Dcp2 and Nst1 Condensates at Diauxic Shift Are Dispersed in 1,6-Hexanediol-Treated Cells but Not Rapidly Disassembled in CHX-Treated Cells

Stress-induced PB accumulation in exponentially growing cells has a couple of typical characteristics, while the physical properties of PBs induced in the cells of post-exponential phases have not been elucidated in detail. The stress-induced PBs of exponentially growing cells immediately diminish when the stress disappears, suggesting that the PBs of the log phase are dynamic cellular structures whose generation and degeneration are highly reversible [24]. The dynamics of PB accumulation and dismantlement closely correlate with the physical properties of LLPS, which is also presented by 1,6-hexanediol treatment [25]. 1,6-hexanediol was reported to function in eradicating the nuclear pore permeability barrier by interfering hydrophobic interactions in the pores and is generally used to interfere with the integrity of reversible condensates with liquid-like properties [26,27]. In budding yeast cells, treatment of 5–10% 1,6-hexanediol for 30 min can impede PB integrity but cannot disperse irreversible amyloids [25]. To examine whether the accumulated PBs in the post-exponential growth-phase cells have similar physical properties to PBs accumulated in the stressed log phase cells, the PB marker protein Dcp2 on a diauxic shift was explored when examining Nst1. We first examined the dynamics of Dcp2 condensates in cells in a diauxic shift by switching the cells to a fresh glucose-sufficient medium. The w303a strains of Figure 1B at the diauxic shift were transferred to fresh glucose-sufficient medium and incubated for 15 min for observation. The Dcp2-EGFP condensates of diauxic shift disappeared in the microscopic images by switching cells to a glucose-sufficient SD medium (Figure 2A), whereas the protein expression levels of Dcp2-EGFP analyzed using Western blotting were stably maintained (Appendix A). These observations suggested that Dcp2-EGFP condensates in the diauxic shift were rapidly dispersed throughout the cytoplasm when the glucose-deficient stress was eliminated. When we also examined these cells in the diauxic shift after treatment with 10% 1,6-hexanediol for 30 min, the Dcp2-EGFP condensates formed in the diauxic shift were dispersed by 1,6-hexanediol, similar to the dispersion of Dcp2 by supplementing glucose (Figure 2A). Analogous to Dcp2, the accumulated Nst1 condensates in the diauxic shift were reduced rapidly in a fresh glucose-sufficient medium and by 1,6-hexanediol treatment (Figure 2A), although the protein levels of endogenous EGFP-Nst1 remained constant, as evaluated using Western blot analysis (Appendix A). These observations demonstrated that PBs in the diauxic shift cells have liquid-like properties similar to those of exponentially growing cells under stress, and strongly support the association of Nst1 with PBs.

As stress-induced PB accumulation is reduced when CHX is treated, the polysome-free RNA is recognized as an indispensable factor in the stressed-induced PB condensation of exponentially growing cells [28]. Thus, we examined whether PB accumulation in cells at the diauxic shift is affected by the decrease in polysome-free RNA levels in the cytoplasm with CHX. Cells expressing Dcp2-EGFP or Nst1-EGFP at the diauxic shift were treated with 100 μg/mL CHX for 10 min. Cells in the glucose-deprived log phase (OD_600_ < 0.5) expressing Dcp2-EGFP, which were prepared as in Figure 1B, were also treated with 100 μg/mL CHX for 10 min as a control. As previously reported, the Dcp2-EGFP condensates induced by glucose deprivation for 10 min were dispersed after CHX treatment (Figure 2B). In contrast, Nst1-EGFP condensates and Dcp2-EGFP accumulation in cells of diauxic shift did not disappear after CHX treatment (Figure 2B). In systematic puncta analysis, which was examined by measuring the maximum pixel intensity of each puncta using FIJI ImageJ, the reduction of Dcp2-EGFP accumulation in glucose-deprived cells was clearly detected after CHX treatment, but the accumulation of Dcp2 or Nst1 at the diauxic shift was not alleviated after CHX treatment (Figure 2C), as shown in Figure 2B. The maximal pixel intensities of the puncta in these cells were not reduced but were significantly enhanced after CHX treatment (Figure 2C). Altogether, these observations in Figure 2 strongly suggest that the accumulated Dcp2 condensates in cells at the diauxic shift are reversible aggregates with liquid-like properties, but their condensations are less affected by changes in the amount of polysome-free RNA in the cytoplasm. Similar characteristics of the Nst1-EGFP condensates to those of the Dcp2-EGFP condensates also support Nst1 as a PB-associated protein in the post-exponential growth phase.

### 2.3. Overexpressed Nst1 Drives the Condensation of PB Components by Self-Condensation

The results shown in Figure 1 and Figure 2 suggest that Nst1 is strongly associated with PBs in cells at diauxic shifts. Then, we tried to clarify whether Nst1 is a PB-associated ‘client’ protein or functions as a ‘scaffold’ protein inducing the condensation of other PB components [5]. Since the deletion or overexpression of ‘scaffold’ proteins residing in cellular condensates is reported to enhance or diminish the condensation of other PB components [5], we evaluated PB formation in NST1-deleted or -overexpressed cells.

EGFP-tagged PB components in exponentially growing cells were observed as miniscule puncta dispersed in the cytoplasm (Figure 1B and Figure 3A). When these cells are exposed to various types of stress, they rapidly turn into intensified puncta. In addition, the miniscule condensates of PBs in exponentially growing cells were chronically intensified over the culturing time from the exponential to post-exponential growth phases. Because both processes of PB accumulation from small to large bodies are not discrete but continuous, it is challenging to determine the degree of PB accumulation. In previous studies, PB accumulation was quantified by the number of cells containing intensified puncta [29]. Instead, to accurately measure the extent of accumulation during the process, we applied a systematic quantification method by image processing utilizing FIJI ImageJ (https://imagej.net/software/fiji/, accessed on 7 January 2022) with the BaSiC plugin [30]. All puncta in the cells of the control and experimental groups were randomly captured during the same exposure time, after focusing by phase microscopy. After correcting the images to normalize the background using the BaSiC plugin, the pixels of the top 0.05% signal intensity value were segmented to analyze the puncta. We detected punctate accumulation in two ways. First, the puncta density was investigated by measuring the maximum intensity of the puncta of each strain. The analysis of the maximum intensity values of puncta was a qualitative investigation of puncta representing the degree of puncta density, and the number of cells containing puncta was not considered. To complement this analysis, we quantified Dcp2-EGFP puncta by measuring the integrated density per cell, which validated the quantity of punctate accumulation per cell. These two types of measurements were applied to all subsequent analyses, and the method is described in detail in the Section 4.

Using these puncta analysis tools, we examined condensates of Dcp2-EGFP endogenously expressed both in BY4741 wild-type cells (YSK3485) and ∆*nst1* cells (YSK3506) in log phase (OD_600_ < 0.5) and at diauxic shift (OD_600_ 2–3). The accumulation of PBs at the diauxic shift was highly increased in both wild-type and ∆*nst1* cells compared to that of the cells in the log phase. When the accumulation of PBs at the diauxic shift between the wild-type and ∆*nst1* cells was compared, no significant difference was detected in both quantitative and qualitative puncta analyses (Appendix A). These data concur with previous reports and the recent model of cellular condensation, which notes that deletion mutants of PB components may not affect the accumulation of other components [19]. In addition, these data represent the redundancy of the PB accumulation pathways for diauxic shifts.

Next, we investigated whether overexpression of Nst1 induces condensation of the PB component by examining endogenous Dcp2-EGFP accumulation in Nst1-overexpressed cells with a vector-only control. The overexpression of Nst1 was guided by the galactose-inducible (*gal*) promoter in plasmid pMW20 (U). However, considering that PB formation might be affected by glucose depletion during galactose-inducible ectopic expression of Nst1, we first monitored the formation of Dcp2-EGFP puncta in only pMW20 (U) vector-transformed cells during the galactose-induced expression process as a control. As expected, the PB assembly increased significantly during incubation in raffinose for 3 h and was reduced after additional incubation for 3 h in galactose, demonstrating the background effect of the raffinose medium on PB formation (Appendix A). Since the addition of up to 0.1% glucose to raffinose reduces starvation-related toxicity and retains the overexpression efficiency [31], we checked whether a raffinose medium with 0.1% glucose alleviates background Dcp2 puncta formation, compared to a raffinose and raffinose/galactose medium without glucose. As expected, the background effect of the raffinose medium was alleviated by using a raffinose medium containing 0.1% glucose (Appendix A). When discovering the effect of enhancing PB accumulation by overexpression of Nst1 or Edc3, shown in Figure 4, we further confirmed we could detect the enhanced PB accumulation of Nst1 and Edc3 by overexpression using a raffinose medium with 0.1% glucose (Appendix A). Thus, a raffinose medium with 0.1% glucose followed by 2% galactose was used in subsequent Nst1 overexpression experiments.

To verify that ectopic overexpression of Nst1 affects the condensation of PB components, the ectopic expression of Nst1 was induced in the wild-type BY4741 strains that expressed endogenous Dcp2-EGFP (a PB marker, YSK3578), Dhh1-EGFP (a PB marker, YSK3564), and Pab1-EGFP (a SG marker, YSK3569), and EGFP puncta accumulation was monitored in each strain. Each strain was transformed using only the vector, then also used as a negative control for comparison with Nst1 overexpression. Surprisingly, the accumulation of Dcp2-EGFP and Dhh1-EGFP condensates was significantly increased in Nst1-overexpressed cells, whereas Pab1-EGFP did not induce any condensation (Figure 3A). Both the integrated density per cell and the average maximum intensity value of Dcp2-EGFP and Dhh1-EGFP puncta in Nst1-overexpressed cells were significantly higher than those in the vector control (Figure 3B,C). On the other hand, the intensity of Pab1-EGFP did not change significantly by Nst1 overexpression compared to that of the vector control (Figure 3B,C). We also confirmed the expression of Dcp2-EGFP, Dhh1-EGFP, and Pab1-EGFP in Nst1-overexpressed cells using Western blot analysis to confirm that Nst1 overexpression did not affect the expression levels of Dcp2, Dhh1, and Pab1. The protein levels of Dcp2-EGFP, Dhh1-EGFP, and Pab1-EGFP were unaffected by Nst1 overexpression (Appendix A), indicating that the increased accumulation of Dcp2-EGFP and Dhh1-EGFP puncta upon Nst1 overexpression was not due to changes in their cellular concentration.

To understand how overexpression of Nst1 enhances Dcp2 and Dhh1 puncta generation, EGFP-tagged Nst1 under the *gal* promoter in pMW20 was transformed into cells with endogenous Dcp2 (YSK3485), Dhh1 (YSK3484), and Pab1 (YSK3530) tagged with mKate2. Upon the induction of overexpression, the overexpressed EGFP-tagged Nst1 was observed as bright clear puncta, which co-localized with mKate2-tagged Dcp2 and Dhh1 puncta (Figure 3D). However, the dynamics of mKate2 tagged-Pab1 were not affected by Nst1 overexpression (Figure 3D). These results confirm that Nst1 condensates are associated with PBs, suggesting that Nst1 has the intrinsic potential to enhance PB accumulation by self-condensation.

### 2.4. Nst1 Overexpression Induces Dcp2 Accumulation by Self-Condensation, Similar to Edc3, Whereas Dcp2, Dhh1, and Pub1 Overexpression Does Not

Next, we investigated whether the overexpression of other known PB components can induce PB formation, as observed with Nst1 in Figure 3. Thus, we examined whether ectopic expression of other PB components could increase the accumulation of Dcp2-EGFP puncta, such as Nst1. Endogenous Dcp2-EGFP puncta in wild-type cells (YSK3485) were monitored after ectopic expression of Edc3, Dhh1, and Dcp2 (PB markers) in pMW20 under the control of the *gal* promoter. Nst1 and SG marker Pub1 in pMW20 under the *gal* promoter were also overexpressed as controls. Overexpression of Edc3 induced puncta formation as the overexpression of Nst1 (Figure 4A). Overexpression of Edc3 also increased the condensation of Dcp2-EGFP puncta as much as Nst1 overexpression in both the analysis of the maximum intensity of puncta and the integrated density per cell (Figure 4B,C). On the other hand, overexpression of Dcp2, Dhh1, and Pub1 did not enhance puncta formation compared to that in the vector-only control (Figure 4A–C). In the case of Dcp2, the endogenous Dcp2-EGFP puncta rather decreased upon its overexpression, compared to that in the vector control, suggesting a dominant negative role of ectopically overexpressed Dcp2 in endogenous EGFP-tagged Dcp2 accumulation (Figure 4A–C). We also checked the expression level of endogenous Dcp2 in these cells and found that it was not much changed by the overexpression of Edc3, Nst1, Dcp2, Dhh1, and Pub1 (Appendix A). These observations strongly support the possibility that Dcp2 accumulation enhanced by overexpression of Edc3 is functionally relevant to PB accumulation, similar to that of Nst1.

As only overexpression of Edc3 and Nst1 led to PB accumulation, we tested the self-aggregation capacity of Edc3, Dcp2, and Dhh1, as in Nst1. We also examined the co-localization of overexpressed EGFP-tagged Nst1, Edc3, Dhh1, and Dcp2 with endogenous mKate2-tagged Dcp2. Each EGFP-tagged protein was overexpressed in the wild-type, where the endogenous chromosomal *DCP2* was tagged with mKate2 (YSK3578). As expected from the results shown in Figure 4A–C, only EGFP-tagged Edc3 overexpression yielded bright and discrete puncta among the overexpression of tested PB components, as EGFP-Nst1 was overexpressed (Figure 4D). Both overexpressed EGFP-Nst1 and EGFP-Edc3 co-localized with endogenously expressed mKate2-Dcp2 (Figure 4D). In contrast, only a minor portion of the overexpressed EGFP-Dcp2 and EGFP-Dhh1 appeared to associate with endogenous mKate2-tagged Dcp2, and a major portion of each protein was dispersed in the cytoplasm (Figure 4D). These observations suggest that the enhanced Dcp2 condensation induced by Nst1 overexpression is functionally similar to Dcp2 accumulation upon Edc3 overexpression. 

### 2.5. Condensates Induced by Overexpressed Nst1 and Edc3, Less Dependent on Polysome-Free mRNA, Exhibit a Liquid-Like Property

So far, we have observed that Nst1 is associated with PBs, and its self-condensation induces PB accumulation, similar to the overexpression of Edc3. As polysome-free mRNA is sufficient for endogenous PB accumulation induced by stress [28], we investigated whether polysome-free RNA is an indispensable factor in Nst1 and Edc3 self-condensation or PB accumulation induced by Nst1 and Edc3 self-condensation. After Edc3, Nst1, Dhh1, and vector-only control were overexpressed in cells expressing endogenous Dcp2-EGFP, the cells were treated with 100 μg/mL CHX for 10 min to reduce polysome-free RNA concentration in the cytoplasm. We observed that the accumulation of Dcp2-EGFP condensates induced by Edc3 and Nst1 overexpression persisted in the presence of CHX (Figure 5A). The maximum intensity of Dcp2-EGFP puncta in both Edc3 and Nst1-overexpressing cells did not decrease even after CHX treatment, suggesting that Nst1 and Edc3 have intrinsic potential to induce self-condensation as well as condensation of other PB components, which is less dependent on polysome-free mRNA (Figure 5B,C). The endogenous Dcp2-EGFP protein levels shown in the Western blots confirmed that the Dcp2-EGFP expression level was not changed by the overexpression of Nst1 and Edc3 (Appendix A). The Dcp2 protein levels of Edc3, Nst1, and Dhh1-overexpressed cells were stably maintained before and after treatment with CHX, indicating that CHX treatment did not affect the Dcp2 protein concentration in cells (Appendix A). These results strongly suggest that the intrinsic properties of Nst1 and Edc3 are critical for the self-condensation and accumulation of other PB component condensates.

The observation that Nst1, upon overexpression, formed condensates that co-localized with PB as Edc3 prompted us to investigate whether Nst1 accumulation has a liquid-like property. We treated cells containing condensates formed by Nst1 overexpression with hexanediol, as shown in Figure 2A. *EGFP-NST1* and *EGFP-EDC3* under the control of the *gal* promoter were induced for 3 h in the wild-type and further incubated for 30 min with or without 10% hexanediol. The condensates of EGFP-Nst1 and EGFP-Edc3 formed upon overexpression were obvious in the hexanediol-untreated control (Figure 5D). Conversely, the condensates of EGFP-Nst1 and EGFP-Edc3 dissolved after 30 min of treatment with hexanediol, suggesting that the Nst1 condensates have liquid-like properties similar to those of Edc3 condensates (Figure 5D). EGFP-tagged Nst1 and Edc3 protein levels in hexanediol-treated and -untreated cells were monitored using Western blot analysis. Overexpression of Nst1 and Edc3 did not decrease in hexanediol-treated cells, confirming that the dispersed EGFP signal was due to the disruption of LLPS, not due to protein degradation (Appendix A).

We also confirmed the liquid-like properties of Dcp2 condensates driven by Nst1 and Edc3 overexpression. Nst1 and Edc3 were overexpressed in w303a cells expressing endogenous Dcp2-EGFP, and these cells were treated with CHX for 10 min to exclude polysome-free RNA-driven condensates. These cells were then treated with 10% hexanediol for 30 min to verify the physical properties of the Dcp2 condensates. Dcp2-EGFP condensates induced by Nst1 and Edc3 overexpression and sustained by CHX treatment were dispersed after treatment with hexanediol, indicating that the liquid-like property of Dcp2-EGFP condensates was driven by Nst1 and Edc3 overexpression (Figure 5E).

### 2.6. Accumulation of Dcp2-EGFP Condensates by Nst1 Overexpression Is Decreased in ∆edc3 lsm4∆C as Compared to That in Wild-Type Cells

Previous studies have demonstrated that Edc3 and the C-terminal prion-like domain of Lsm4 are essential for PB assembly in response to glucose deprivation in the log phase [14,15]. In the presence of glucose deprivation, the ∆*edc3 lsm4*∆*C* cells do not produce Dcp2 condensates, while wild-type cells rapidly accumulate Dcp2 condensates [14]. To further confirm that Nst1 overexpression induces PB formation as Edc3, we examined the endogenous Dcp2-EGFP accumulation by the ectopic expression of Nst1 and compared it with that in the ectopic expression of Edc3 and the vector control in the cells of ∆*edc3 lsm4*∆*C*. When overexpression of Nst1 and Edc3 was induced in ∆*edc3 lsm4*∆*C* (yRP2340) expressing endogenous Dcp2-EGFP, Dcp2-EGFP puncta were clearly detected in both the Nst1- and Edc3-overexpressed ∆*edc3 lsm4*∆*C* cells but not in ∆*edc3 lsm4*∆*C* cells overexpressing the vector control (Figure 6A). The maximum intensities of puncta as well as the integrated density of EGFP-Dcp2 per cell was significantly increased in both Nst1 and Edc3-overexpressing cells, compared to that in the cells of the vector-only overexpressed control (Figure 6B,C). The Dcp2-EGFP protein expression was confirmed using the Western blot analysis after overexpression of Nst1 and Edc3, demonstrating that the overexpression of Nst1 and Edc3 did not affect Dcp2-EGFP protein concentration in these cells, although we could detect the accumulation of Dcp2-EGFP (Appendix A). These observations confirmed the function of overexpressed Nst1 and Edc3 on PB assembly and showed that Nst1, when overexpressed, has the potential to interact and accumulate PB components.

However, upon Nst1 overexpression, we recognized that Dcp2-EGFP condensates of ∆*edc3 lsm4*∆*C* cells were not as strong as those of wild-type cells, shown in Figure 4 and Figure 6D, though the accumulation of Dcp2-EGFP condensates caused by ectopic expression of Nst1 in ∆*edc3 lsm4*∆*C* cells was obvious compared to that of the vector control (Figure 6A–C). To compare the accumulation of Dcp2-EGFP condensates induced by Nst1 overexpression between wild-type cells and ∆*edc3 lsm4*∆*C* cells, we utilized the relative response ratio (RRR) that is a parameter usually used in normalizing the luciferase reporter gene assay (https://www.promega.kr/resources/pubhub/bioluminescent-reporter-assay-normalization/, accessed on 7 January 2022). We calculated the RRR of the integrated density value of Dcp2-EGFP puncta per cell upon Nst1 overexpression, using the value of the vector control as a negative control and the value of Edc3 overexpression as a positive control. The RRR of negative control and positive control was 0 and 1, respectively, and the RRR of integrated density value on Nst1 overexpression was calculated as −0.0883 that indicated the accumulation of Dcp2-EGFP condensates induced by Nst1 overexpression in ∆*edc3 lsm4*∆*C* cells was reduced, comparing to the accumulation of Dcp2-EGFP condensates induced by Nst1 overexpression in wild-type cells (Figure 6E). These data suggest that overexpressed Nst1 enhance the accumulation of Dcp2-EGFP condensates majorly through Edc3 and, at the same time, suggest that this pathway through Edc3 might not be the only way that Dcp2-EGFP accumulation is induced by Nst1.

## 3. Discussion

In this study, we showed that Nst1, a novel component of PBs, is densely associated with PBs during the post-exponential growth phases and has scaffold-like properties. We also demonstrated that PBs chronically accumulated in post-exponential growth phases might be distinct from the abrupt formation of PBs in response to stress during exponential growth. It is well known that the malfunction of intracellular condensation or de-condensation due to chronic aging is closely related to the generation of irreversible amyloids, resulting in severe physiological disorders [32,33,34]. The demonstration of novel components that are potentially responsible for inducing condensates, especially in post-exponential growth phases in budding yeast, may provide clues for solving the mechanisms underlying cellular condensation in chronic aging and in neurodegenerative diseases.

### 3.1. Nst1 Is a Specific Component of PBs in the Cells of Post-Exponential Phases

Nst1 is a large protein with 1240 amino acids, including diverse sequence elements such as a highly charged central region. We previously reported that Nst1 functions as an adaptor-like protein that cross-links mitogen-activated protein (MAP) kinase pathways in response to stress [17]. We investigated the function of Nst1 in PB accumulation because we observed that the antiproliferative phenotype of its overexpression was suppressed by well-known components of PBs, such as Psp2 and Pop2 (Appendix A). Although we are studying the function of Nst1 in PB formation, another group has recently reported it as a novel PB component [29]. When we recognized that the endogenous EGFP-tagged Nst1 is strongly self-condensed and co-localized with the Dcp2 and Edc3 condensates in the post-exponential growth phases, we expected that the endogenous EGFP-tagged Nst1 in exponentially growing cells would be observed as puncta similar to those of cells exposed to glucose deprivation. However, unlike previously known PB components such as Dcp2, Dhh1, and Edc3, Nst1 did not appear as intensive puncta by abrupt glucose deprivation, as its condensates were observed in the diauxic shift and stationary phase cells (Figure 1B, Appendix A). The observations in this study showed that Nst1 is associated with PBs and contributes to PB accumulation specifically in post-mitotic diauxic and stationary phase cells but is less effective in glucose-deprived exponentially growing cells. They also imply that the mechanisms underlying PB formation in abrupt glucose deprivation and in chronic aging by culture time, might be distinct.

### 3.2. PBs Chronically Accumulated in Post-Exponentially Growing Phases Might Be Different Compared to PBs Abruptly Formed in Response to Stress in Exponentially Growing Cells

One of the strong evidences for the distinct mechanisms of PB formation in rapid stress responses and in chronical aging of post-exponential phases is the accumulation of condensates of Dcp2-EGFP in the ∆*edc3 lsm4*∆*C* mutant [15]. The ∆*edc3 lsm4*∆*C* mutant has been reported as the most effective suppressor of PB accumulation in exponentially growing cells [14,15]. Dcp2 condensates were not induced in the stress condition of ∆*edc3 lsm4*∆*C* mutant in exponential growth [14,15]. However, formation of Dcp2 condensation was not suppressed even by the most potent ∆*edc3 lsm4*∆*C* double mutant in stationary phase cells [15], suggesting that a different subset of inducers possibly enhances the accumulation of PB condensates depending on various cellular circumstances. Consistently, we showed in this study that intense Dcp2-EGFP condensates in cells at the diauxic shift were not decreased even after CHX treatment, while Dcp2-EGFP condensates formed by glucose deprivation in the exponential growth phase were disassembled by CHX, although both condensates have liquid-like properties (Figure 2). Our results imply that polysome-free RNA is essential to initiate PB assembly in glucose-deprived cells of exponential growth [28], but proteins might play essential roles to induce chronic accumulation of PBs more than mRNAs do. These observations altogether strongly support the notion that the PPI interaction network, as well as the inducer of PB formation such as scaffold(s), would be distinct depending on various cellular circumstances.

### 3.3. Nst1 Has a Scaffold-Like Property Similar to Edc3

The overexpression system is an effective application for demonstrating a novel scaffold and its role in condensation. However, it also has limitations in that any protein may form condensates when its concentration exceeds the threshold concentration for saturation [35,36,37]. Therefore, the self-condensation phenomenon of *gal*-induced Nst1 alone may not confirm the function of Nst1 in the formation of PBs. In this study, we resolved this issue by examining the overexpression of other essential PB components such as Dcp2, Dhh1, and Edc3 using the same *gal*-inducible promoter.

Overexpression of EGFP-tagged Lsm4, a representative PB component with a prion-like domain (polyQ motif) in its C-terminal region, was previously reported to drive self-condensation [25]. In comparison to the stress-responsive endogenous Lsm4-EGFP puncta dissipated by CHX, puncta generated by *gal*-induced EGFP-Lsm4 were not dispersed by CHX [25], implying that the puncta induced by overexpressed Lsm4 are not identical to those of the stress-derived endogenous PBs. Although the puncta upon overexpression do not have physical properties identical to those of native PBs, these observations provide important molecular information about Lsm4 that Lsm4 has the intrinsic potential to enable self-oligomerization.

Our results showed that overexpressed EGFP-Edc3 appeared as bright clear puncta, consistent with a previous report that Edc3 harbors the Yjef-N domain, which induces Edc3 self-oligomerization [38]. We observed that EGFP-Nst1 overexpression formed bright clear puncta that were not dissipated by CHX. Moreover, overexpression of Nst1 induced not only Dcp2-EGFP condensates, but also condensates of Dhh1 (Figure 3A–C), Xrn1, and Edc3 (data not shown). These observations strongly suggest that Nst1 has a sequence element that induces self-oligomerization similar to Lsm4 or Edc3, although Nst1 does not have a previously known prion-like domain. The RNA-binding domain and intrinsic disordered regions (IDRs) are common features of scaffold proteins that induce condensate formation [8]. In recent models, the multivalency of proteins consisting of condensates induces promiscuous interactions that drive condensate accumulation [8]. Nst1, a large protein of 1240 amino acids with no predicted three-dimensional structure and containing a large proportion of predicted IDR and highly charged amino acid (K, R, E, G) repeats, is expected to have self-oligomerizing domain(s) and IDR(s) as Edc3 and Lsm4. Based on our data, further studies of Nst1 domains are required to investigate the logic for understanding the mechanism of self-oligomerization or multivalency, which would provide influential information for understanding the grammar of protein condensation.

### 3.4. Significant Portion of Overexpressed Nst1 Induces PB Accumulation via Edc3 and C-Terminal Domain of Lsm4

Unlike other physiological phenomena, biomolecular condensation is not quantitatively regulated. For example, deletion mutants of PB components, such as Xrn1, Dcp1, and Dcp2, did not decrease the condensation of other PB components, but some enhanced the condensation of other PB components [19]. We also observed no significant difference in the accumulation of Dcp2-EGFP condensates between the wild-type and ∆*nst1* cells of the diauxic shift. Recent models have explained this discrepancy with the multivalency and oligomerization properties of the components in cellular condensates. That is, when one interaction is blocked, other competitive interactions can be reinforced because of the multivalency of the component. We expect that there is still the potential for enhanced condensation of certain other components by Nst1 deletion in this model. These data also represent the functional redundancy of PB accumulation pathways for diauxic shifts.

We observed that overexpression of Edc3 could induce the accumulation of Dcp2, and the accumulation of Dcp2 condensates in Nst1-overexpressed cells was similar to that in Edc3-overexpressed cells. These data raise the question of whether the multivalency or oligomerizing properties of Nst1 are mediated by Edc3 or not. If the overexpressed Nst1 could oligomerize and multivalently interact with other PB components dependently on Edc3, the Dcp2-EGFP condensates would not be accumulated in ∆*edc3 lsm4*∆*C* cells. In contrast, the Dcp2-EGFP condensates could be accumulated even in ∆*edc3 lsm4*∆*C* cells if the overexpressed Nst1 could oligomerize and multivalently interact with other PB components independently of Edc3. In our results, the accumulation of Dcp2 in Nst1-overexpressed ∆*edc3 lsm4*∆*C* cells was enhanced compared to that in the vector-only control (Figure 6A–C). However, the degree of accumulation of Dcp2 produced by overexpressed Nst1 in ∆*edc3 lsm4*∆*C* mutant cells was much lower than that of Dcp2 puncta produced by Nst1 overexpression in w303a wild-type cells, when compared with Dcp2 accumulation by Edc3 overexpression both in wild-type and ∆*edc3 lsm4*∆*C* mutant cells (Figure 6D,E). This relative reduction of Dcp2-EGFP condensates in ∆*edc3 lsm4*∆*C* cells compared to that in wild-type cells strongly suggest the possibility that Nst1 induces the accumulation of PB components somehow through the Edc3 and Lsm4 C-terminal domain in part. The investigation of the functional relevance of Nst1, Edc3 and Lsm4 as scaffolds for condensation remains as future work.

## 4. Materials and Methods

### 4.1. Yeast Strains, Plasmids, and Culture

The *S. cerevisiae* strains used in this study are listed in Table 1 with their genotypes. The strains were constructed on the BY4741 or w303a wild-type background by integrating templates from the polymerase chain reaction (PCR) toolbox at the 3′ end of each reading frame in each endogenous locus through a PCR-based homologous recombination [39]. All of the constructed strains were verified using a PCR of the integrated locus and Western blotting.

All plasmids used in this study were constructed in *pMW20(U)- P_GAL_(GAL10-1) or*, *pMW20(U)- P_GAL_(GAL10-1) -EGFP* as described in Table 2. Each gene was inserted into pMW20(*U*) containing pCEN-*P_GAL_* (a CEN-based vector with the *GAL10-1* promoter) to observe the overexpression phenotypes.

To induce overexpression from the *GAL10-1(P_GAL_)* promoter, the yeast strains were cultured in a synthetic complete (SC) media containing 2% glucose at 25 °C. The cells were cultured to an optical density at 600 nm (OD_600_) ≤ 0.5 for logarithmic phase growth. Cells at the diauxic shift were cultured to an optical density at 600 nm (OD_600_) of 2–3. To induce overexpression under the *gal* promoter, cells in logarithmic phase were primarily cultured in a SC-U (uracil) + 2% glucose medium to an OD_600_ of 0.5 and harvested. The cells were washed three times with a SC-U + 2% raffinose + 0.1% glucose medium, diluted to half of its concentration, and cultured for an additional 3 h in SC-U + 2% raffinose + 0.1% glucose. Then, 20% galactose stock was added to the culture to adjust the final galactose concentration to 2%, and the cells were further incubated for 3 h for induction before collection.

### 4.2. Genetic Screening for the Suppressor of Overexpressed Nst1

Ninety-two gene deletion mutants from the YKO library (Saccharomyces Genome Deletion Project, http://www-sequence.stanford.edu/group/yeast_deletion_project/, accessed on 7 January 2022) were transformed with pMW20(U)-pGAL or pMW20(U)-pGAL-NST1. The transformed cells were cultured in a SC-U + 2% glucose medium up to the exponential growth phase (OD600 0.5) and harvested. The cells were washed with a SC-U + 2% raffinose medium three times, diluted to half its concentration, and cultured to an OD600 of 1 in SC-U + 2% raffinose. The same amount of culture was prepared, followed by serial 10-fold dilutions in SC-U + 2% raffinose. Then, 20 µL of each dilution was spotted onto SC-U + 2% glucose and SC-U + 2% raffinose + 2% galactose agar plates. The plates were incubated at 25 °C.

### 4.3. Hexanediol and CHX Treatment

The yeast pGAL induction procedure has been previously described. To treat the cells with hexanediol, the *gal*-induced cells were washed three times with a SC medium containing 10% hexanediol + 2% raffinose + 2% galactose + 0.1% glucose and incubated for 1 h. For the CHX treatment, the *gal*-induced cells were incubated with 100 μg/mL CHX for 10 min.

### 4.4. Wide-Field Fluorescence Microscopy of Yeast Cells and Image Analysis

Fluorescence-labelled proteins were visualized using an Axioplan2 microscope (Carl Zeiss, Jena, Germany) with a 100× Plan-Neofluar oil immersion objective. Images were acquired using an Axiocam CCD camera and the Axio Vision software (Carl Zeiss). To quantify the PB condensation, all images were obtained using the same optics, filters, zoom settings, and fluorescence exposure times throughout the study and were analyzed using FIJI ImageJ. Sequential analyses (A–D) were performed, as described below. (A) Image obtainment: A phase contrast image of the cells was first captured for focusing before fluorescence imaging. The same exposure time (e.g., 1 s) was applied to all the fluorescence images being compared. All of the cells were captured once to avoid signal bleaching. (B) Background correction: Images were corrected using the BaSiC plugin in FIJI ImageJ, and automatic regularization was applied with dark-field and flat-field correction using predefined shading profiles [30]. (C) Puncta segmentation: Images were transformed to 8-bit. The top 0.05% intensity pixels in each image were segmented for particle analysis. (D) Puncta measurement: The maximal intensity value of each segmented puncta was projected onto the plot. The sum of each punctum integrated density was divided by the number of cells as the integrated density per cell.

### 4.5. Relative Response Ratio (RRR)



$$\mathrm{RRR}\;=\;\frac{\frac{{I{}^{\circ}(P_{\mathrm{GAL}}\;-\;\mathrm{NST}1)}_{\Delta \mathrm{edc}3{}^{\circ}\mathrm{lsm}4\Delta C}}{{I(P_{\mathrm{GAL}}\;-\;\mathrm{NST}1)}_{\mathrm{wildtype}}}\;-\;\frac{{I{}^{\circ}(P_{\mathrm{GAL}})}_{\Delta \mathrm{edc}3{}^{\circ}\mathrm{lsm}4\Delta C}}{{I{}^{\circ}(P_{\mathrm{GAL}})}_{\mathrm{wildtype}}}}{\frac{{I{}^{\circ}(P_{\mathrm{GAL}}\;-\;\mathrm{EDC}3)}_{\Delta \mathrm{edc}3{}^{\circ}\mathrm{lsm}4\Delta C}}{{I(P_{\mathrm{GAL}}\;-\;\mathrm{EDC}3)}_{\mathrm{wildtype}}}\;-\;\frac{{I{}^{\circ}(P_{\mathrm{GAL}})}_{\Delta \mathrm{edc}3{}^{\circ}\mathrm{lsm}4\Delta C}}{{I{}^{\circ}(P_{\mathrm{GAL}})}_{\mathrm{wildtype}}}}$$



*I* = Integrated density per cell

*RRR* of ∆*edc3 lsm4*∆C mutant on *Nst*1 overexpression was calculated using the integrated density value of Dcp2-EGFP puncta per cell of each strain. Wildtype cells expressing endogenous Dcp2-EGFP upon vector control and Edc3 overexpression were used as a negative control (*RRR* = 0) and positive control (*RRR* = 1), respectively.

### 4.6. Western Blotting

Protein samples were separated by sodium dodecyl sulfate polyacrylamide gel electrophoresis (SDS-PAGE) and then transferred to polyvinylidene difluoride membranes (Merck Millipore, Burlington, MA, USA). Anti-EGFP antibody (600-101-215 ROCKLAND, Limerick, PA, USA) was used for Western blotting, and the same blot was probed with anti-Tub1 (T5168, Sigma, St.Louis, MO, USA) as a positive control. HRP-conjugated anti-goat (705-035-003, Jackson Immune Research) and anti-mouse (sc-2005, Santa Cruz Biotechnology, Dallas, TX, USA) antibodies were used as secondary antibodies for detecting EGFP and anti-Tub1, respectively.

### 4.7. Statistical Analysis

Detailed statistics, including the mean values and standard deviations, are indicated in each figure legend. Statistical analyses were performed using GraphPad Prism version 9.2.0 (GraphPad Software, Inc., La Jolla, CA, USA). We used the Student’s *t* test or nested t test to assess statistically significant differences. *p* < 0.05 (*), *p* < 0.01 (**), *p* < 0.001 (***), and *p* < 0.0001 (****) indicate statistical significance compared with the control. *p* > 0.05 indicates statistical non-significance (n. s.).

## Figures and Tables

**Figure 1 ijms-23-02501-f001:**
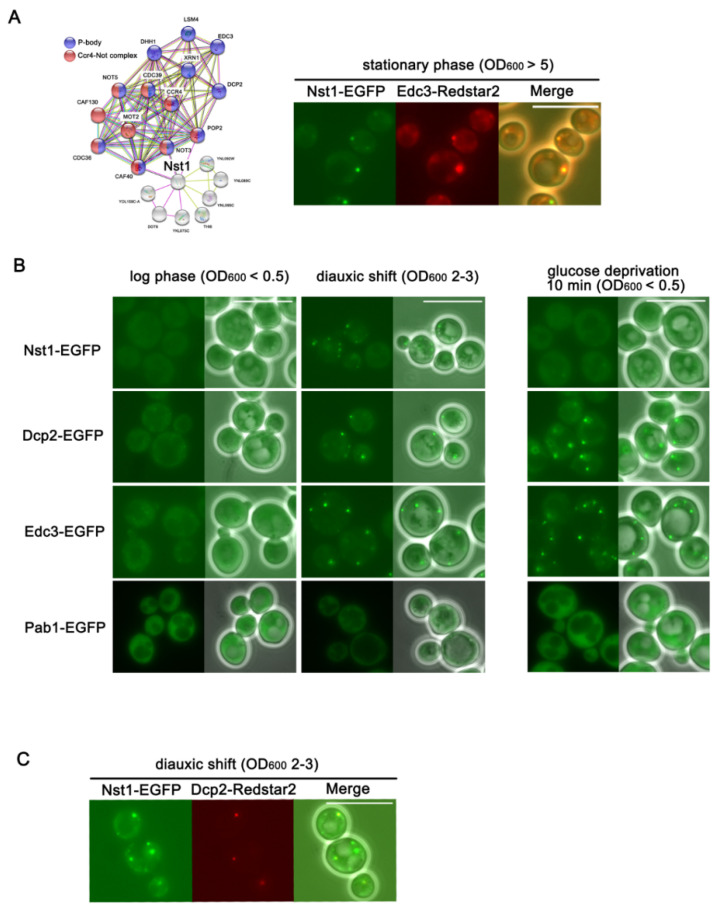
Nst1 is densely associated with PBs in post-exponential growth phase cells of *Saccharomyces cerevisiae.* (**A**) The Nst1 protein–protein interaction network with PB components including Ccr4-Not complex was visualized using STRING (functional enrichment analysis of protein–protein interaction networks, https://string-db.org/, accessed on 7 January 2022). The purple dots represent previously known PB components, and the red dots indicate components of the Ccr4-Not complex. Co-localization of the endogenous EGFP-tagged Nst1 and Redstar2-tagged Edc3 (YSK 3520) in w303a cells of stationary phase (OD_600_ > 5). Scale bar, 10 μm. (**B**) Fluorescence microscopy of w303a cells endogenously expressing Nst1-EGFP, Dcp2-EGFP, Edc3-EGFP, and Pab1-EGFP in the exponential growth (logarithmic phase), diauxic shift, and glucose-depleted stress condition. YSK3524, YSK3527, YSK3528, and YSK3525 strains, primarily expressing EGFP-fused Nst1, Dcp2, Edc3, and Pab1 respectively, were observed at log phase (OD_600_ < 0.5) and diauxic shift (OD_600_ of 2–3). Log phase cells (OD_600_ < 0.5) were washed three times with glucose-depleted synthetic medium and additionally incubated for 10 min to induce glucose deprivation. Scale bar, 10 μm. (**C**) Co-localization of the endogenous EGFP-tagged Nst1 and Redstar2-tagged Dcp2 (YSK3509) in w303a cells of diauxic shift (OD_600_ 2–3). Scale bar, 10 μm.

**Figure 2 ijms-23-02501-f002:**
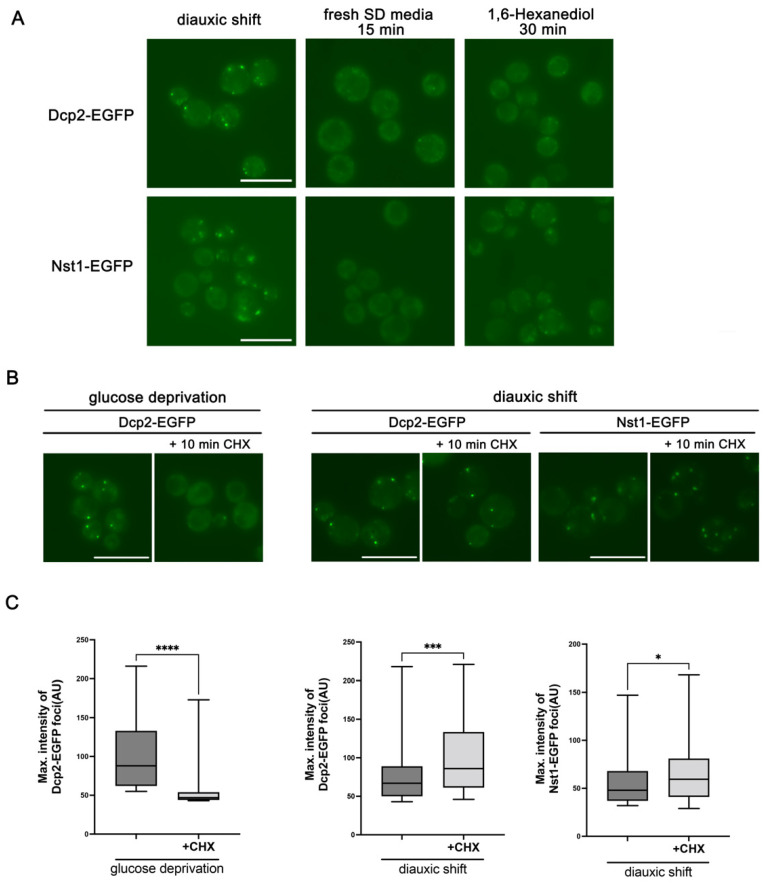
Nst1 condensates in diauxic phase cells are dispersed by 1,6-hexanediol but not rapidly disassembled by CHX, similar to Dcp2 condensates. Endogenous Nst1-EGFP (YSK3524) and Dcp2-EGFP (YSK3527)-expressing w303a cells were cultured up to a diauxic shift (OD_600_ of 2–3) and observed using fluorescence microscopy. (**A**) Conversion of Nst1-EGFP and Dcp2-EGFP foci of diauxic shift cells was monitored after these cells were transferred to a fresh SD medium for 15 min or to a medium containing 10% 1,6-hexanediol for 30 min. Scale bar, 10 μm. (**B**) The accumulation of Dcp2-EGFP and Nst1-EGFP condensates in cells on diauxic shift was examined before and after treatment with 100 μg/mL CHX. For comparison, cells endogenously expressing Dcp2-EGFP in the exponential growth phase were washed three times and incubated in a glucose-depleted medium for 10 min to induce glucose deprivation before being treated with 100 μg/mL CHX for 10 min. Scale bar, 10 μm. (**C**) Quantification of Nst1 and Dcp2 condensate accumulation in (**B**). The foci of Dcp2-EGFP and Nst1-EGFP were segmented, and the maximum pixel intensity values of Nst1-EGFP and Dcp2-EGFP foci were measured and analyzed using FIJI ImageJ. The methods for the segmentation and quantification of condensates are described in detail in the Section 4. Condensates in more than 60 cells under each condition were analyzed. Error bars show mean ± standard error of the mean (SEM). Statistical significance was determined using Student’s *t* test (* *p* < 0.05, *** *p* < 0.001, **** *p* < 0.0001).

**Figure 3 ijms-23-02501-f003:**
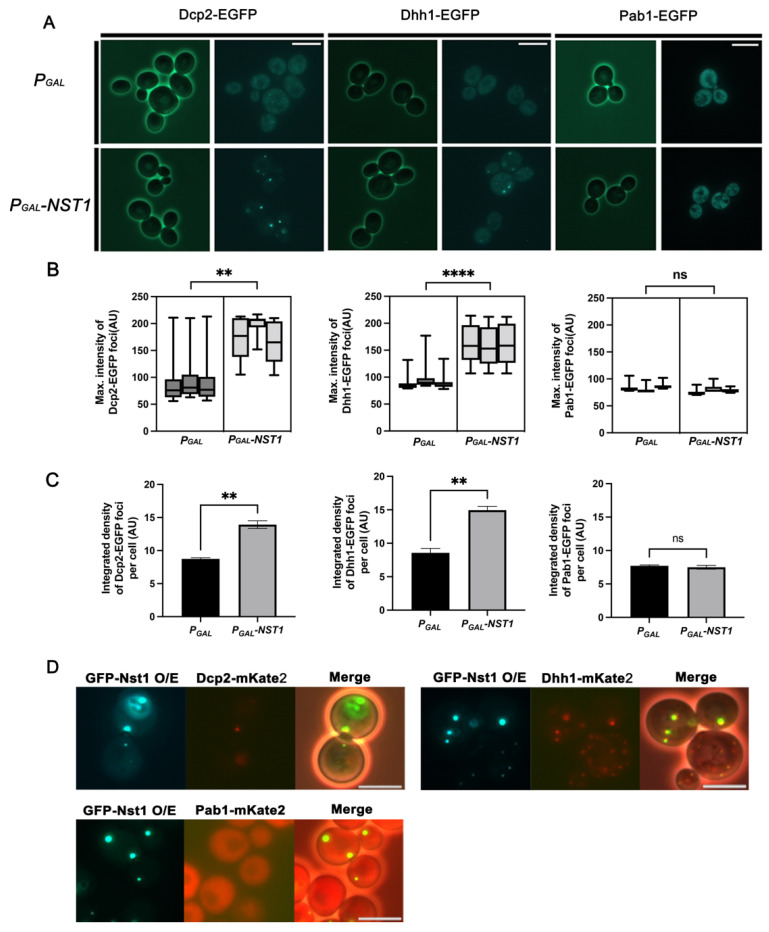
Overexpressed Nst1 drives condensation of PB components by self-condensation. (**A**–**C**) The wild-type strains, whose chromosomal *DCP2* (YSK3485), *DHH1* (YSK3484), and *PAB1* (YSK3530) were respectively tagged with *EGFP*, were transformed with *pMW20-pGAL* and *pMW20-pGAL-NST1*. Each strain cultured in a SD-Uracil (2% glucose) medium was transferred to a 2% raffinose + 0.1% glucose medium for 3 h, and galactose was added to a final 2% concentration. The cells were additionally cultured for 3.5 h for induction of galactose-inducible(*gal*) promoter. (**A**) Fluorescence microscopy of the endogenous EGFP-tagged Dcp2, Dhh1, and Pab1, when Nst1 or vector-only control was overexpressed. Scale bar, 5 μm. (**B**,**C**) Measurements of puncta accumulation of the endogenous EGFP-tagged Dcp2, Dhh1, and Pab1 in the Nst1-overexpressed wild-type cells of (**A**). More than 300 cells of each strain were monitored for one independent experimental analysis, and every experiment was repeated three times. The segmentation and quantification methods are described in detail in the Section 4. Error bar shows mean values ± standard error of the mean (SEM). Statistical significance was determined using a Student’s *t* test (** *p* < 0.01, **** *p* < 0.0001). (**B**) Monitoring maximum signal intensities of the condensation of PB and SG components by Nst1 overexpression in (**A**). Maximum pixel intensity values of Dcp2-EGFP, Dhh1-EGFP, and Pab1-EGFP puncta of vector-only control (*P_GAL_*) and Nst1-overexpressed (*P_GAL_-NST1*) cells were measured and analyzed using FIJI ImageJ. (**C**) Quantitative analysis of Dcp2-EGFP, Dhh1-EGFP, and Pab1-EGFP puncta per cell by Nst1 overexpression in (**A**). Total integrated density of each Dcp2-EGFP, Dhh1-EGFP, and Pab1-EGFP segmented puncta in vector-only control and Nst1-overexpressed cells was determined using FIJI ImageJ. (**D**) Wild-type strains expressing endogenous Dcp2-mKate2 (YSK3578), Dhh1-mKate2 (YSK3654), and Pab1-mKate2 (YSK3569) were transformed with *pMW20-P_GAL_-GFP-NST1.* Overexpressed EGFP-tagged Nst1 as well as endogenous mKate2-tagged Dcp2, Dhh1, and Pab1 were visualized in each strain using fluorescence microscopy. Scale bar, 5 μm.

**Figure 4 ijms-23-02501-f004:**
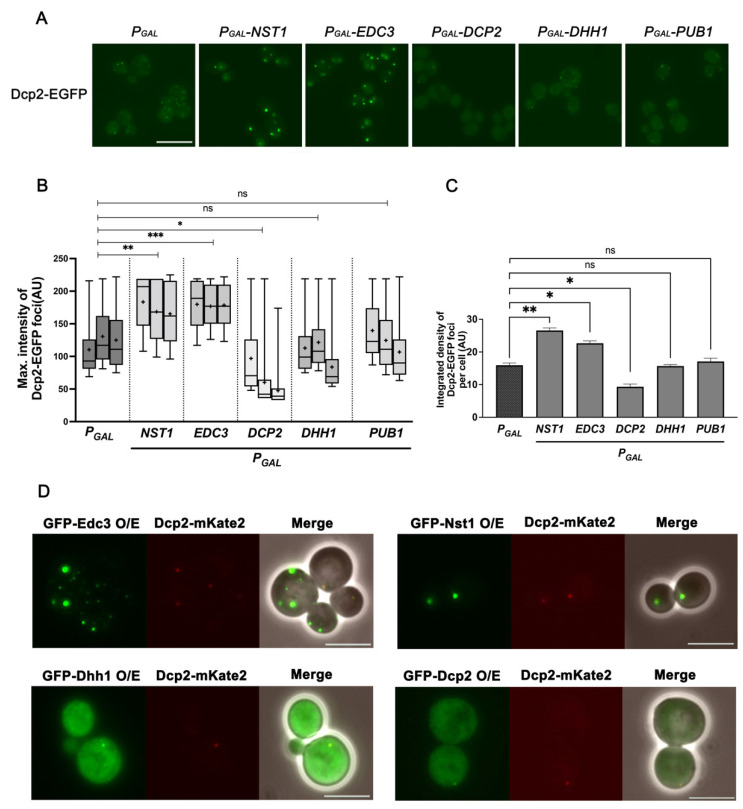
Nst1 overexpression induces Dcp2 accumulation by self-condensation similar to Edc3, while overexpression of Dcp2, Dhh1, and Pub1 does not. (**A**–**C**) Dcp2 accumulation in the cells respectively overexpressing Nst1, PB components Edc3, Dcp2, and Dhh1, and a SG component Pub1. The wild-type cells, whose chromosomal *DCP2* was tagged with *EGFP* (YSK3485), were transformed with *pMW20-pGAL* (vector control), *pMW20-pGAL-NST1*, *pMW20-pGAL-EDC3*, *pMW20-pGAL-DCP2*, *pMW20-pGAL-DHH1*, and *pMW20-pGAL-PUB1*. Cells cultured in a SD-U (2% glucose) medium were transferred to a 2% raffinose + 0.1% glucose medium for 3 h, and galactose was added to a final 2% concentration to induce the expression for 3.5 h. (**A**) Fluorescence microscopy of the endogenous Dcp2-EGFP in cells respectively overexpressing Nst1, Edc3, Dcp2, Dhh1, and Pub1. Scale bar, 10 μm. (**B**,**C**) Measurements of Dcp2-EGFP puncta induced by the overexpression of Nst1, Edc3, Dcp2, Dhh1, and Pub1 of (**A**). Approximately, 200 cells of each strain were monitored for one independent analysis, and every experiment was repeated three times. Pixels of top 0.05% Dcp2-EGFP signal intensity were segmented for the analysis. The segmentation and quantification methods applied to the images are described in the Section 4. (**B**) Monitoring maximum signal intensities of Dcp2-EGFP puncta by overexpression of Nst1, Edc3, Dcp2, Dhh1, and Pub1. The maximal intensity of segmented puncta was plotted on the *y*-axis and presented using a boxplot. The error bar shows mean values ± standard error of the mean (SEM). Statistical significance was determined using a nested t test (* *p* < 0.05, ** *p* < 0.01, *** *p* < 0.001). (**C**) Quantitative analysis of Dcp2-EGFP puncta produced per cell by Nst1, Edc3, Dcp2, Dhh1, and Pub1 overexpression. The total integrated density of Dcp2-EGFP condensates in the vector-only control and Nst1, Edc3, Dcp2, Dhh1, and Pub1-overexpressed cells was determined. The error bar shows mean values ± standard error of the mean (SEM). Statistical significance was determined using a Student’s *t* test (* *p* < 0.05, ** *p* < 0.01, *** *p* < 0.001). (**D**) Fluorescence microscopy of ectopically overexpressed GFP-Edc3, GFP-Dhh1, GFP-Dcp2, and GFP-Nst1 in the cells whose endogenous Dcp2 was tagged with mKate2 (YSK3578). Scale bar, 5 μm.

**Figure 5 ijms-23-02501-f005:**
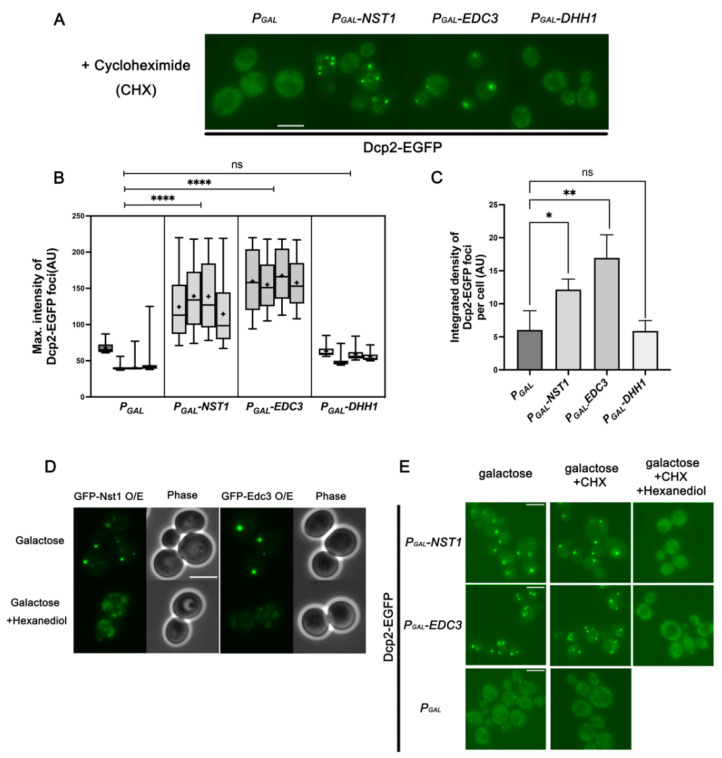
Dcp2-EGFP condensation induced by Nst1 and Edc3 overexpression is maintained in CHX-treated cells but has a liquid-like property. (**A**–**C**) Dcp2-EGFP condensation after the induced expression of vector control, Nst1, Edc3, and Dhh1. Cells, whose chromosomal *DCP2* was tagged with *EGFP* (YSK3485), were transformed with *pMW20-pGAL* (vector control), *pMW20-pGAL-NST1*, *pMW20-pGAL-EDC3*, and *pMW20-pGAL-DHH1.* After raffinose-galactose induction, each strain was treated with 100 µg/mL CHX, incubated for 10 min, and observed. (**A**) Fluorescence microscopy of Dcp2-EGFP after CHX treatment in the cells expressing ectopic Nst1, Edc3, and Dhh1. Scale bar, 5 μm. (**B**,**C**) More than 200 cells of each strain were monitored for one independent experimental analysis, and every experiment was repeated four times. Pixels of top 0.05% Dcp2-EGFP signal intensity were segmented for the analysis. The segmentation and quantification methods applied to the images are described in the Section 4. (**B**) Monitoring maximum signal intensities of Dcp2-EGFP puncta in (**A**). The maximal intensity of segmented puncta was plotted on the *y*-axis and presented using a boxplot. The error bar shows mean values ± standard error of the mean (SEM). Statistical significance was determined using a nested t test (* *p* < 0.05, ** *p* < 0.01, **** *p* < 0.0001). (**C**) Quantitative analysis of Dcp2-EGFP puncta per cell of (**A**). Total integrated density of Dcp2-EGFP condensates in the vector-only control and Nst1-, Edc3-, and Dhh1-overexpressed cells was determined. The error bar shows mean values ± standard error of the mean (SEM). Statistical significance was determined using a Student’s *t* test (* *p* < 0.05, ** *p* < 0.01, **** *p* < 0.0001). (**D**) Fluorescence microscopy of GFP-Nst1 and GFP-Edc3 condensates induced in the wild-type cells by their overexpression (top) and after further treatment with 10% 1,6-hexanediol (bottom). Overexpression of GFP-Nst1 and GFP-Edc3 was induced in BY4741 wild-type cells (YSK3483) transformed with *P_GAL_-GFP*, *P_GAL_-GFP-NST1*, and *P_GAL_-GFP-EDC3*, respectively, using raffinose-galactose induction procedures described in the Section 4. Cells containing GFP-Nst1 and GFP-Edc3 condensates were treated with 10% hexanediol for 1 h. Scale bar, 5 μm. (**E**) Fluorescence microscopy of endogenous Dcp2-EGFP condensation induced by the overexpression of Nst1 and Edc3 (left) and by CHX treatment following their overexpression (middle) and by 1,6-hexanediol treatment following CHX treatment after their overexpression (right). The wild-type cells whose chromosomal *DCP2* was tagged with *EGFP* (YSK3485) were transformed with *P_GAL_*, *P_GAL_-NST1*, and *P_GAL_-EDC3*, and their overexpression was induced by raffinose-galactose as described in the Section 4. After induction, each transformed strain was treated with 100 µg/mL CHX for 10 min or 100 µg/mL CHX followed by 10% 1,6-hexanediol for 30 min. Scale bar, 5 μm.

**Figure 6 ijms-23-02501-f006:**
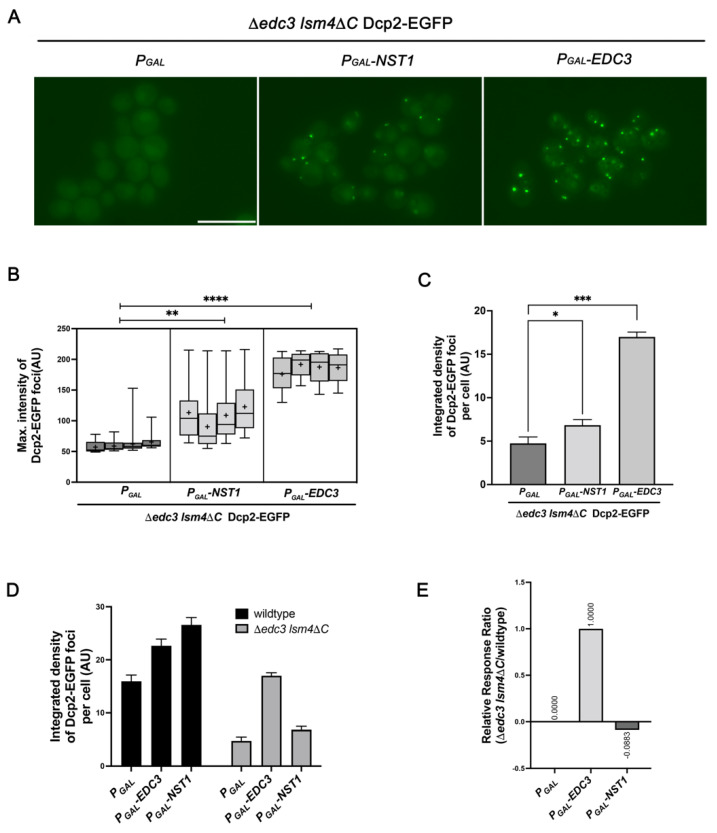
Overexpression of Edc3 and Nst1 induces endogenous Dcp2-EGFP accumulation in ∆*edc3 lsm4*∆*C.* Dcp2-EGFP condensation in the ∆*edc3 lsm4*∆*C* strain after induction of expression of the vector control, Nst1, and Edc3. The *edc3* and *lsm4* C-terminal domain double deletion mutant (∆*edc3 lsm4*∆*C)*, whose chromosomal *DCP2* was tagged with *EGFP* (yRP2340), was transformed with *pMW20-pGAL* (vector control), *pMW20-pGAL-NST1*, and *pMW20-pGAL-EDC3.* Each strain was observed and analyzed after raffinose-galactose induction, as described in the Section 4. (**A**) Fluorescence microscopy of Dcp2-EGFP in yRP2340 cells ectopically expressing the vector control, Nst1, and Edc3. Scale bar, 10 μm. (**B**,**C**) More than 300 cells of each strain were monitored for one independent analysis, and every experiment was repeated four times. Pixels with the highest 0.05% Dcp2-EGFP signal intensity were segmented for analysis. The segmentation and quantification methods applied to the images are described in the Section 4. (**B**) Monitoring the maximum signal intensities of the Dcp2-EGFP puncta in (**A**). The maximal intensity of the segmented puncta was plotted on the *y*-axis and presented in the box plot. The error bars show mean ± standard error of the mean (SEM). Statistical significance was determined using a nested t test (* *p* < 0.05, ** *p* < 0.01, *** *p* < 0.001, **** *p* < 0.0001). (**C**) Quantitative analysis of Dcp2-EGFP puncta production per cell of (**A**). The total integrated density of Dcp2-EGFP condensates in the vector-only control, and Nst1 and Edc3-overexpressed cells was determined. The error bars show mean ± standard error of the mean (SEM). Statistical significance was determined using Student’s *t* test (* *p* < 0.05, ** *p* < 0.01, *** *p*< 0.001, **** *p* < 0.0001). (**D**) Bar graph representing the values of integrated density per cell for Dcp2-EGFP condensates in the vector control, and Nst1 and Edc3-overexpressed cells of ∆*edc3 lsm4*∆*C* and w303a wild-type strains. The mean values of integrated density per cell measured in Figure 4C of w303a wild-type cells and in Figure 6C of ∆*edc3 lsm4*∆*C* cells were used to calculate the relative response ratio (RRR) of the Dcp2 condensation upon Nst1 overexpression in ∆*edc3 lsm4*∆*C* cells. The error bar shows mean ± standard error of the mean (SEM). (**E**) The RRR of Dcp2-EGFP integrated density per cell in ∆*edc3 lsm4*∆*C* mutant cells upon Nst1 overexpression. The formula for the RRR calculation is described in the Section 4.

**Table 1 ijms-23-02501-t001:** List of the yeast strains used in this study.

Strain Name	Genotype	Source
yRP2340	*MAT leu2-3.112 trp1 ura3-52 his4-539 cup1::LEU2/PGK1pG/MFA2pG lsm4 C:: NEO edc3:: NEO Dcp2GFP(NEO)*	R. Parker
YSK3483	BY4741 *MATa his3Δ1 leu2Δ0 met15Δ0 ura3Δ0*	R. Parker
YSK3484	BY4741 except as *DHH1-EGFP:HIS3MX6*	this study
YSK3485	BY4741 except as *DCP2-EGFP:HIS3MX6*	this study
YSK3506	BY4741 except as ∆*nst1::KANMX DCP2-GFP::HIS3MX*	this study
YSK3509	W303a except as *Nst1-EGFP::KANMX6 Dcp2-Redstar2::NATNT2*	this study
YSK3519	W303a except as *EDC3-EGFP::HIS3MX6*	this study
YSK3520	W303a except as *NST1-EGFP::KANMX6 EDC3-Redstar2::NATNT2*	this study
YSK3524	W303a except as *NST1-EGFP::KANMX4*	this study
YSK3525	W303a except as *PAB1-EGFP::HIS3MX6*	this study
YSK3527	W303a except as *DCP2-EGFP::HIS3MX6*	this study
YSK3530	BY4741 except as *PAB1-EGFP::HIS3MX6*	this study
YSK3549	W303a *leu2-3*,*112 trp1-1 can1-100 ura3-1 ade2-1 his3-11*,*15*	S. Elledge
YSK3564	BY4741 except as *DHH1-mKate2-sphis5+*	this study
YSK3569	BY4741 except as *PAB1-mKate2-sphis5+*	this study
YSK3578	BY4741 except as *DCP2-mKate2-sphis5+*	this study
YSK3642	BY4741 except as *NST1-EGFP::HIS3MX6 PAB1-mKate2::CaURA3*	this study
YSK3644	BY4741 except as *NST1-EGFP::HIS3MX6 DCP2-mKate2::CaURA3*	this study

**Table 2 ijms-23-02501-t002:** List of the plasmids used in this study.

Plasmid	Source
*pMW20(U)-P_GAL_*	K. Song
*pMW20(U)-P_GAL_-EGFP*	K. Song
*pMW20(U)-P_GAL_-DCP2*	this study
*pMW20(U)-P_GAL_*-*DHH1*	this study
*pMW20(U)-P_GAL_*-*EDC3*	this study
*pMW20(U)-P_GAL_*-*NST1*	this study
*pMW20(U)-P_GAL_*-*PUB1*	this study
*pMW20(U)-P_GAL_*-*EGFP-DCP2*	this study
*pMW20(U)-P_GAL_*-*EGFP-DHH1*	this study
*pMW20(U)-P_GAL_*-*EGFP-EDC3*	this study
*pMW20(U)-P_GAL_*-*EGFP-NST1*	this study
*pMW20(U)-P_GAL_*-*EGFP-PUB1*	this study

## Data Availability

Not applicable.

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
