# Peer review of "Nst1, Densely Associated to P-Body in the Post-Exponential Phases of Saccharomyces cerevisiae, Shows an Intrinsic Potential of Producing Liquid-Like Condensates of P-Body Components in Cells"

_ijms, 2022, doi:10.3390/ijms23052501_

Round 1

Reviewer 1 Report

Dear edithor, dear authors,

The paper of Yoon Jeong Choi and Kiwon Son is very interesting, revealing new proprieties of the Nst1 protein.

The authors proved that Nst1 is a specific component in the P Bodies in the Saccharomyces cerevisiae cells in post-exponential growth phases with a scaffold-like property similar to Edc3.

They studied the behavior of the cells in deletion and overexpression conditions of the NST1 gene. It was observed that overexpression of NST1 leads to the condensation of P Bodies components, and in the same time induces Dcp2 accumulation by self-condensation, similar to Edc3. The condensates induced by the overexpression of Nst1 and Edc3 exhibit liquid-like proprieties. The data presented suggest the possibility that Nst1 induces the accumulation of PB components through Edc3 and Lsm4 C-terminal domain in part.

The work is well presented and structured. The experiments have been very well designed and carried out and the conclusions achieved on the basis of the experimental data are good justified and very useful. The methods used in this study are specific and the experimental is very well explained.

The paper could be read by a native speaker to ensure the correct use of English in a few points, however, there are no major grammatical errors that would make the text difficult to comprehend.

I recommend publishing the manuscript in its present form.

Author Response

There was no specific request for the revision.

Reviewer 2 Report

In this manuscript, Choi and Song present a very interesting study on the role of Nst1 in producing liquid-like condensates of P-body (PBs) components in Saccharomyces cerevisiae cells growing in the post-exponential phases, by showing that Nst1 is more densely associated with PBs in post-exponentially growing phases from diauxic shift to stationary phase than during glucose deprivation of exponentially growing cells. The study is well substantiated, presenting robust and sound data. The manuscript is very well written, but there are some issues that the authors need to address before the manuscript can be accepted for publication.

  • The rationale behind the genetic screening for the suppressor of overexpressed Nst1 is not fully clear. What were the 90 KO mutants used in the screen? On what basis were they selected from YKO library? What were the results of the screening? All these should be included in a dedicated paragraph in the Results section.
  • The authors should indicate why they used W303-A in some experiments and BY4741 in others.
  • In Table 1, please indicate the provenance of the wild-type strains.
  • In Table 2, please specify the characteristics of plasmid pMW20-Pgal, and also its provenance.
  • Section 4.2: please indicate the provenance of the YKO library.
  • In Δedc3lsm4ΔC, leave a space between the two gene names.

Author Response

Reviewer #2

  • The rationale behind the genetic screening for the suppressor of overexpressed Nst1 is not fully clear. What were the 90 KO mutants used in the screen? On what basis were they selected from YKO library? What were the results of the screening?

As written in the manuscript lines 92-108, genome-wide in vivo protein-protein interaction (PPI) study by Tarassov et al. showed that Nst1 interacts functionally with the components of the Ccr4-Not complex associated with PBs. Since we discovered that the Nst1-overexpressed cells show growth retardation compared to the vector control, we performed suppressor screening for Nst1 to confirm the functional association between Nst1 and PBs. We screened the deletion mutants of the genes related to the function of PB or SG that could suppress the growth retardation phenotype of the Nst1-overexpressed cells (Supplementary Figure S1A). The 92 deletion mutants in the yeast knock-out (YKO) library (Saccharomyces Genome Deletion Project, http://www-sequence.stanford.edu/group/yeast_deletion_project/), which were categorized as genes related to PB or SG in the cerevisiae genome database (SGD, https://www.yeastgenome.org), were used for the screening (Supplementary Figure S1B). As a result, we identified PSP2, LSM12, DCS1, TIF4631, and POP2 as suppressor genes whose deletion suppressed the growth retardation phenotype induced by Nst1 overexpression. Considering that ectopic expression of Psp2 recovers PB assembly in glucose-deprived ∆edc3 lsm4∆C cells and that Pop2 is a component of the Ccr4-Not deadenylase complex, the functional relevance of Nst1 to PB components was suggested. Based on these functional interactions with the previously-known PB components, we investigated the function of Nst1 in PB formation.

All these were included and improved in the Results section (lines 92-109) of the revised manuscript.

  • The authors should indicate why they used W303-A in some experiments and BY4741 in others.

We observed the same phenomena of Nst1-EGFP both in w303a and BY4741 strain. However, the endogenous expression level of Nst1 was too low to get clear image of Nst1-EGFP with microscopy in exponential growing cells. We found that the Nst1-EGFP signal in w303a is slightly better to be detected than that in BY4741. BY4741 strain usually has smaller size than w303a, thus the w303a strain is advantageous when take images.

  • In Table 1, please indicate the provenance of the wild-type strains.(o)
  • In Table 2, please specify the characteristics of plasmid pMW20-Pgal, and also its provenance.(o)
  • Section 4.2: please indicate the provenance of the YKO library.(o)
  • In Δedc3lsm4ΔC, leave a space between the two gene names.

We fixed all minor requests in the revised manuscript.

Reviewer 3 Report

In this manuscript, the authors reported potential functions of Nst1 as a member of P-body. The whole manuscript is well-written and contained valuable results. My minor comments are as follows:

  1. P-body stands for processing body as I read at Abbreviation (line 754). The authored may consider to explain this term in the introduction section.

  1. Page 2, line 60: Generally, ‘Δ’ doesn’t have to be italic.

  1. Page 5, line 129: 5 < OD600 or OD600 > 5?

  1. All six figures have extra ‘Fig x’ symbol at the up-left side. Please remove them.

  1. Pages 11, lines 358-359: Incomplete sentence, please revise.

  1. Page 15, line 462: In the main text, the authors use mL, but here is ml. Please be consistent.

  1. Page 17, line 561: Nst1 is a 1240 amino acid protein. Page 19, lines 628-629: Nst1 is a 1210 amino acid protein. Please confirm.

  1. Page 20, Table 1: minor suggestion, the authors can bring YSK3509, 3519, 3520 to the upper side, and switch 3644 with 3622.

  1. Page 22, line 720: Please come up some sentences to explain the formula.

  1. Page 22, line 746: What does MSIT stand for?

  1. Page 22, lines 748-750: IRB, consent, and data availability cannot be blank.

  1. Page 23, references: ref format isn’t matched with ijms. Please revise.

  1. Ref 19, cerevisiae with a small c; in addition, the name should be italic. Ref 23, 24, 32 and 35, pages are missing.

Author Response

Reviewer #3

  • P-body stands for processing body as I read at Abbreviation (line 754). The authored may consider to explain this term in the introduction section.(o)
  • Page 2, line 60: Generally, ‘Δ’ doesn’t have to be italic.(o)
  • Page 5, line 129: 5 < OD600 or OD600 > 5?(o)
  • All six figures have extra ‘Fig x’ symbol at the up-left side. Please remove them.(o)
  • Pages 11, lines 358-359: Incomplete sentence, please revise(o).
  • Page 15, line 462: In the main text, the authors use mL, but here is ml. Please be consistent.(o)
  • Page 17, line 561: Nst1 is a 1240 amino acid protein. Page 19, lines 628-629: Nst1 is a 1210 amino acid protein. Please confirm.(o)
  • Page 20, Table 1: minor suggestion, the authors can bring YSK3509, 3519, 3520 to the upper side, and switch 3644 with 3622.(o)
  • Page 22, line 720: Please come up some sentences to explain the formula.(o)
  • Page 22, line 746: What does MSIT stand for?(o)
  • Page 22, lines 748-750: IRB, consent, and data availability cannot be blank(o). 
  • Page 23, references: ref format isn’t matched with ijms. Please revise.(o)
  • Ref 19, cerevisiae with a small c; in addition, the name should be italic. Ref 23, 24, 32 and 35, pages are missing.(o)

We fixed all above comments in the revised manuscript. Authors appreciate for the reviewer’s careful reading and suggestion.
